# Simple Equations Pertaining to the Particle Number and Surface Area of Metallic, Polymeric, Lipidic and Vesicular Nanocarriers

**M. R. Mozafari** [1,2,*], **E. Mazaheri** [1] and **K. Dormiani** [3]

1   Australasian Nanoscience and Nanotechnology Initiative, 8054 Monash University LPO, Clayton, VIC 3168, Australia; mazaheri.elaheh@gmail.com
2   Supreme Pharmatech Co. LTD., 399/90-95 Moo 13 Kingkaew Rd. Soi 25/1, T. Rachateva, A. Bangplee, Samutprakan 10540, Thailand
3   Cell Science Research Centre, Department of Molecular Biotechnology, Royan Institute for Biotechnology, ACECR, Isfahan 8165131378, Iran; dormiani@yahoo.com
\*   Correspondence: dr.m.r.mozafari@gmail.com; Tel.: +61-406-284-553

**Abstract:** Introduction: Bioactive encapsulation and drug delivery systems have already found their way to the market as efficient therapeutics to combat infections, viral diseases and different types of cancer. The fields of food fortification, nutraceutical supplementation and cosmeceuticals have also been getting the benefit of encapsulation technologies. Aim: Successful formulation of such therapeutic and nutraceutical compounds requires thorough analysis and assessment of certain characteristics including particle number and surface area without the need to employ sophisticated analytical techniques. Solution: Here we present simple mathematical formulas and equations used in the research and development of drug delivery and controlled release systems employed for bioactive encapsulation and targeting the sites of infection and cancer in vitro and in vivo. Systems covered in this entry include lipidic vesicles, polymeric capsules, metallic particles as well as surfactant- and tocopherol-based micro- and nanocarriers.

**Keywords:** clinical applications; drug targeting; encapsulation; nanoliposome; spherical particles; Theranostics; tocosome

## 1. Introduction

Drug delivery systems emerged around six decades ago with the aim of improving Human and livestock health and well-being. Also known as encapsulation protocols and controlled release systems, they can be broadly categorized into different groups such as lipidic, polymeric, metallic particles, surfactant-based and tocopherol-based carriers (Table 1). The drug delivery systems are very versatile in terms of their ingredients, size, surface characteristics, charge, elasticity, number of coats, layers or bilayers surrounding the drug molecules, encapsulation capacity, release profile and stability/shelf-life. Based on this versatility, drug carriers are being applied in many areas of biological and medical research, diagnosis and therapy as well as a number of different industries (e.g., pharmaceuticals, food, nutrition, cosmetics and skincare) as highly valuable ingredients [1–5]. Formulations of benzoyl peroxide microsponge have been reported to find applications in the treatment of mild to moderate acne [6], while polymeric hydrogel carriers are recently reported to show promise in targeting colon cancer [7,8]. Polymeric nanoparticles [9], niosomes [10] and the recently introduced drug carrier "tocosome" [11,12] are other formulations with potential use to combat cancer, viral and some other health issues.

**Table 1.** Main classes of bioactive encapsulation systems.

| | Category | Examples | Description |
|---|---|---|---|
| 1 | Lipidic systems | Liposome | Bilayer phospholipid vesicles |
| | | Liposphere | Phospholipid monolayer with a solid fat core |
| | | Nanoliposome | Nanometric liposome |
| | | Nanoemulsion | Nanometric oil/water emulsions |
| | | Phytosome | Herbal extracts & natural phospholipid mix |
| 2 | Polymeric carriers | Microsponges | Carrier system composed of porous microspheres |
| | | Hydrogel beads | Polymeric carriers which swell in water |
| | | Polymeric nanoparticles | Nanometric particulate dispersions or solid particles used as bioactive carriers |
| 3 | Surfactant-based | Niosome | Non-ionic surfactant vesicles |
| 4 | Tocopherol-based | Tocosome | Bilayer vesicles composed of tocopheryl derivatives & phospholipids |
| 5 | Metallic particles | Gold particles | |
| | | Copper particles | Colloidal metallic substrates employed as drug delivery systems |
| | | Silver particles | |

Formulation and optimization of potent drug delivery systems necessitate thorough analysis of certain physicochemical and biological characteristics [13,14]. The disposition properties of intravenously injected carrier systems and those applied via other routes of administration are complicated and depend in part on dose, particle size, charge, number of carriers, and total surface area [15–17]. Consequently, parameters such as particle number and surface area must be precisely fine-tuned to ensure successful formulations and their quality assurance. This entry presents simple mathematical formula and equations used in the calculation of the aforementioned parameters along with some available techniques employed to characterize the micro- and nanocarrier systems.

## 2. Particle Number Determination

The tangible number of particles in drug delivery formulations (number concentration, *N*) is of importance for quality assurance, comprehensive physicochemical characterization, and pharmacodynamics [15]. The number of particles in a certain volume of sample, rather than merely their size, could affect their absorption, clearance and disposition [16,18]. The number of particles affects effective uptake of a targeted carrier system by specific cells (e.g., phagocytic cells) and the cumulative drug content of the particles determines their bioactivity and therapeutic efficacy. Knowledge of particle quantity enables the precise assessment of drug concentration in each solid (rigid, e.g., liposomes made of lipids with high phase transition temperature) or liquid (elastic) particle [15]. Concentration of the bioactive agent or its distribution between phases determines if the system is of dissolved or dispersed type, and accordingly, it defines the drug release kinetics and mechanism. Among the techniques used for the quantification of particle number is nanoparticle tracking analysis (NTA), which is able to track and measure particles moving under Brownian motion [17,19]. This high-resolution method is effective in determining the size, size distribution, and concentration of colloidal and particulate drug delivery systems. It can be employed to assess particles and vesicles within the size range of 30–1000 nm [20]. Samples are injected into the special cell of the apparatus and then it is illuminated by laser light (635 nm) that passes through a liquid layer on the optical surface [19,20]. Refraction occurs and the region in which the vesicles or particles are present is illuminated and visualized under microscopy. A charge-coupled device camera records a video (30 frames/sec)

wherein the movement of particles under Brownian motion can be visualized. Special software identifies and tracks the center of each particle throughout the length of the video and relates it to the particle characteristics [21]. Mathematical equations used to calculate the quantity of some examples of carrier systems are described below. Simple equations are given for both solid particles (e.g., metallic particles, polymeric microcapsules and nanocapsules) and soft vesicles (e.g., liposomes, nanoliposomes, lipospheres, solid lipid nanoparticles, niosomes and tocosomes).

### 2.1. Quantification of Number of Metallic Particles

Particles of gold, silver, iron, copper and other metals are popular colloidal substrates employed in various sensor, imaging, and drug delivery applications. They can be synthesized and modified with several different chemical functional groups, which allow them to be conjugated with antibodies, ligands, and other bioactive agents or drugs of interest [22]. Particle number or concentration of metallic particles determines crucial features of the formulations including stability, bioactivity and cytotoxicity. The number of metallic particles ($N_{MP}$) in solution can be calculated from the ratio of the number of initial metal atoms ($N_{ma}$) and the number of metal atoms per one, single metallic particle $N_{ma/smp}$ as described in Equation (1):

$$N_{MP} = N_{ma} \; / \; N_{ma/smp} \tag{1}$$

Hinterwirth and co-workers [23] employed a similar mathematical equation to calculate number concentration of the gold nanoparticles in their formulation. Taking their study as an instance, if initially 55 mL of 1.14 mM Au(III) atoms was used in the construction of particles and the number of metal atoms per single metallic particle is calculated to be 30.89602 (see Equation (2) below), the number of gold nanoparticles in 1 L sample will be:

$$1.14 \times 10^{-3} \text{ mol L}^{-1} \times 0.055 \text{ L} \times N_A \; / \; 30.89602 = 1.2221 \times 10^{18}$$

in which $N_A$ is Avogadro's constant (i.e., 6.02E23). The average number of metal atoms per metallic particle of gold is calculated according to the following equation [24,25]:

$$N_{ma/smp} = \pi \, \rho \, D^3 \; / \; 6M = 30.89602 \, D^3 \tag{2}$$

where $\rho$ is the density for a face-centered cubic (FCC) gold (19.3 g/cm$^3$), D is the average size (31 ± 1.6 nm) and M stands for atomic weight of gold (197 g/mol). The average number of gold atoms per nanoparticle can also be calculated from high-resolution microscopic analysis. Rajakumar et al. [25] reported that images of their synthesized gold nanoparticles depicted particles in the range of 23 to 46 nm, with an average size of 31 ± 1.6 nm (*D*, nm) [25]. Assuming a spherical shape and a uniform face-centered cubic (FCC) structure [26], the average number of gold atoms for each type of nanosphere was calculated by Equation (2) [27,28].

The mathematical approach and the related Equation (1) explained above can be extrapolated to be used for other metallic micro- and nanoparticles including copper [29] and silver [30].

### 2.2. Particle Number of Vesicular Carriers

Vesicular drug-delivery carriers comprise liposomes, nanoliposomes, micelles, tocosomes, niosomes, solid lipid nanoparticles and archaeosomes to name a few [16,31]. Among the vesicular carriers, liposomes (and nanoliposomes) are the most applied encapsulation techniques with the highest number of products approved for Human use on the market. Also known as a bilayer phospholipid vesicle, liposome is a mesomorphic structure mainly composed of lipid, phospholipid and water molecules [32]. The main chemical components of liposomes and nanoliposomes are amphiphilic lipid/phospholipid molecules (Figure 1). They improve the efficacy of pharmaceutical, nutraceutical and other bioactive compounds

by entrapment and release of water-soluble, lipid-soluble and amphipathic materials, as well as targeting the encapsulated drug molecules to particular cells or tissues [33,34]. Vesicular drug carriers, including liposomes, can be prepared in different forms with respect of their number of lamella (phospholipid bilayers) as depicted in Figure 2.

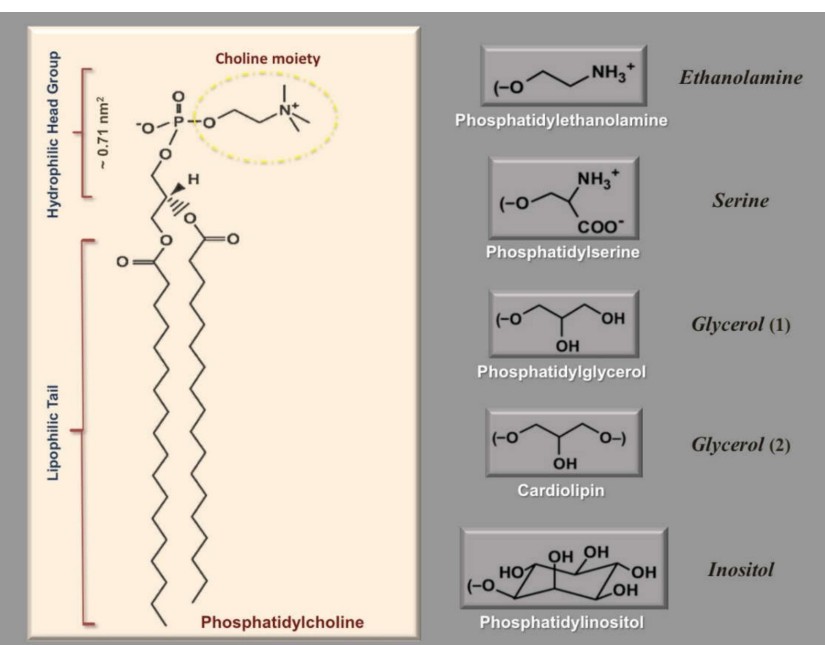

**Figure 1.** Structural formula of phosphatidylcholine molecule and its derivatives/related phospholipids. The chemical moieties on the right panel replace the Choline moiety depicted on the left panel, and hence each phospholipid molecule is named based on its specific moiety on its hydrophilic head group.

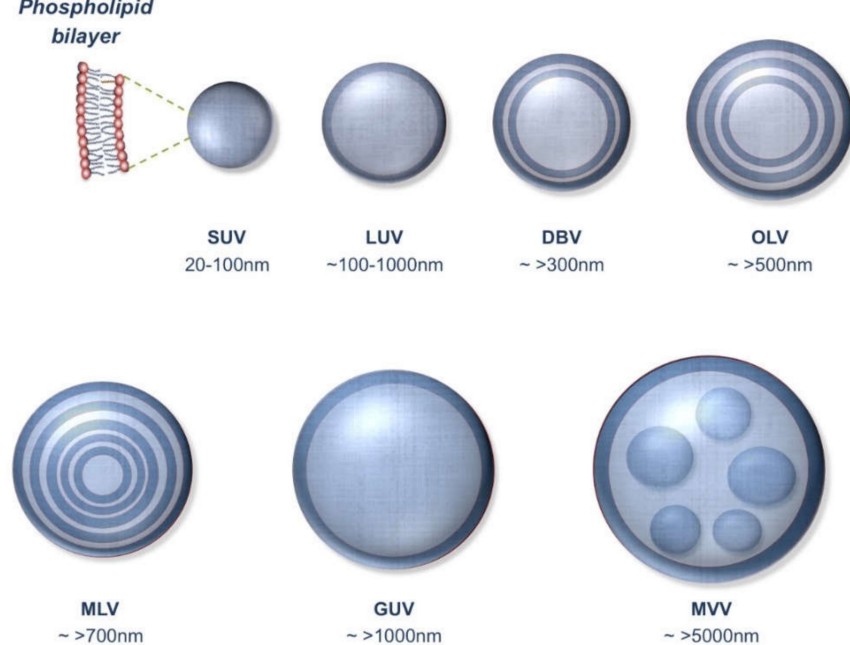

**Figure 2.** Different types of lipid vesicles. SUV: small unilamellar vesicle; LUV: large unilamellar vesicle; DBV: double bilayer vesicle; OLV: oligolamellar vesicle; MLV: multilamellar vesicle; GUV: giant unilamellar vesicle; MVV: multivesicular vesicle.

Currently, there are no validated experimental approaches for the determination of the particle number-concentration of liposomal and nanoliposomal formulations [15]. Here we present a simple mathematical approach to calculate the number of phospholipid vesicles, in the form of unilamellar vesicle, in any certain volume of sample. Once the total concentration of phospholipids, lipids and other ingredients of our vesicles (such as cholesterol, phytosterols, vitamin E, etc.) in the suspending media are known, then the total number of particles per ml can easily be calculated using Equation (3).

$$N_{Ves} = [M_{ing} \times N_A] / [N_{tot} \times 1000] \tag{3}$$

where: $N_{Ves}$ is the number of drug delivery vesicles per milliliter; $M_{ing}$ is the molar concentration of ingredients of the vesicles; $N_A$ is the Avogadro Number (6.02E23); and $N_{tot}$ is the total number of ingredients per vesicle. Equation (3) is the main equation by which particle number of vesicular bioactive carrier systems can be easily calculated once we know $N_{tot}$.

$N_{tot}$ can be calculated using the following equation:

$$N_{tot} = [4 \pi (d/2)E2 + 4 \pi [(d/2)-h)]E2] / a \tag{4}$$

in which: $[4\pi(d/2)^2]$ is the surface area of vesicle's monolayer; *d* is the diameter of the vesicle; *h* is the thickness of the phospholipid bilayer (i.e., ~5 nm); *a* is the phospholipid head group area and E2 is exponent two (to the power 2). The headgroup area of phosphatidylcholine (a generally used ingredient in the manufacture of lipid vesicles, niosomes, tocosomes, etc.) is about 0.71 nm square, as depicted in Figure 1 [35–37]. Accordingly, Equation (4) can be simplified to:

$$N_{tot} = 17.69 \times [(d/2)E2 + (d/2 - 5)E2]$$

in which 17.69 is $4\pi/a$.

As an example the total number of ingredients for a unilamellar formulation with 400 nm mean particle size is:

$$17.69 \times [(400 / 2)^2 + (400 / 2 - 5)^2] = 1,022,088.89 = N_{tot}$$

and using this number as $N_{tot}$ in Equation (3), we will find out the number of vesicles in a milliliter of the prepared sample, assuming molar concentration ($M_{ing}$) of 1 micromole:

$$N_{Ves} = [10^{-6} \times (6.02 \times 10^{23})] / [1,022,088.89 \times 1000] =$$
$$\sim 588,989,868 \text{ vesicles/ml}$$

Exceptional cases for the use of Equation (3) for quantification of particle number of vesicular drug carriers would be multilamellar vesicles (MLV) or multivesicular vesicles (MVV) (see Figure 2). The mathematical equations described above are straightforward means for calculation of particle number. Other approaches mentioned in the literature are complicated and involve multistep calculations. For instance, Pidgeon and Hunt [38] have presented formulas based on the volume of the entrapped water by liposomal vesicles, which involve solving several equations in order to find the estimated particle number.

### 2.3. Particle Number of Polymeric Carriers

A method used to assess the number concentration of polymeric carriers is scanning mobility particle sizer (SMPS) [39,40]. In this method, the formulations are atomized to aerosol droplets that are then dried in a silica-gel diffusion drier. The dry particles flow into the SMPS apparatus, which is composed of the differential mobility analyzer (DMA) unit. DMA provides size information according to the size-dependent electric mobility of the particles. The particles then move into the condensation particle counter (CPC) section, which counts the number of particles in each size group. The combined measurements

result in a highly resolved particle count for the full-size distribution range. Integrating the counts over the full-size range yields the total particle number concentration $N_c$ using Equation (5):

$$N_c = N'_c \times CF \times F_g / R_{ev} \tag{5}$$

in which $N'_c$ is the number of particles per $cm^3$ measured by SMPS, CF is the calibration factor for the particle size, $F_g$ is the flow rate ($cm^3$ $min^{-1}$) of the carrier gas (e.g., nitrogen). $R_{ev}$ is the measured evaporation rate of the sample solution (ml $min^{-1}$) and is calculated by measuring the rate of solution loss from the atomizer compartment for the calibrated gas flow through the atomizer. The calibration factor (CF) is employed to account for the loss of particles in the system.

## 3. Surface Area Determination

When bioactive carriers come in contact with cells and tissues, their external surface is obviously the first contact point, which determines drug pharmacokinetic and pharmacodynamic properties. Knowledge of the surface area of drug delivery systems is of high importance in the assessment of their release and permeation behavior and their stability, as well as their interaction with the target cells and tissues, and hence their cytotoxicity profile. Once we know the number of our spherical or near-spherical particles/vesicles ($N$) and the mean particle size of the formulation ($d$) we can calculate the surface area ($S$) employing Equation (6):

$$S = N \times 4\pi (d/2)E2 \tag{6}$$

in which [$4\pi (d/2)E2$] is the surface area of one single spherical or semispherical particle. Equation (6) can be applied for both solid particles (e.g., metallic particles, polymeric capsules) and soft vesicular systems (e.g., tocosomes, liposomes, nanoliposomes, liposheres, solid lipid nanoparticles, vesicular lipid gels, niosomes and archaeosomes).

While Equation (6) can be used universally to measure surface area of all type of spherical drug carriers, quantification of particle number requires specific equations for different carrier systems based on their ingredients (and as a result their physicochemical properties) as explained throughout this article and simplified in Figure 3.

| Carrier Type | Ingredients | Equation | Morphology |
|---|---|---|---|
| Metallic | gold, silver, copper | $N_{MP} = N_{ma} / N_{ma/smp}$ | |
| Polymeric | poly(lactic-co-glycolic acid), hypromellose, polyethyleneglycol | $N_c = N'_c \times CF \times F_g / R_{ev}$ | |
| Lipidic | lipids, phospholipids | $N_{Ves} = [M_{ing} \times N_A] / [N_{tot} \times 1000]$ | |
| Vesicular (Niosome) | alkyl ethers, spans, tweens | $N_{Ves} = [M_{ing} \times N_A] / [N_{tot} \times 1000]$ | |
| Vesicular (Tocosome) | tocopherylphosphate, di-tocopherylphosphate | $N_{Ves} = [M_{ing} \times N_A] / [N_{tot} \times 1000]$ | |

**Figure 3.** Simple equations for the quantification of particle number of major groups of bioactive carrier systems, generally used ingredients for their manufacture and microscopic image of each class of carrier system.

## 4. Conclusions

Calculations of the particle number and surface area towards formulation of the drug delivery/encapsulation systems is of importance for quality assurance, comprehensive physicochemical characterization, safety, efficacy and pharmacodynamics. Some calculation methods that have been previously employed are limited because they rely on several assumptions and are not applicable for certain formulations of drug carriers. In this entry, we presented simple mathematical equations for the calculation of particle size and surface area of a whole range of bioactive encapsulation systems. Employing data on these characteristics of solid and soft drug carriers will assist in the optimization of formulations intended for pharmaceutical, nutraceutical and skincare applications.

**Author Contributions:** All authors contributed equally to the compilation and editing of the text and tables, as well as graphical designs of figures of this article. All authors have read and agreed to the published version of the manuscript.

**Funding:** This research received no external funding.

**Institutional Review Board Statement:** Not applicable.

**Informed Consent Statement:** Not applicable.

**Data Availability Statement:** https://www.researchgate.net/post/What-is-the-best-way-to-calculate-the-number-of-liposome-nanoparticles-in-a-solution.

**Conflicts of Interest:** The authors declare no conflict of interest.

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
