# Peer review of "Simple Equations Pertaining to the Particle Number and Surface Area of Metallic, Polymeric, Lipidic and Vesicular Nanocarriers"

_scipharm, doi:10.3390/scipharm89020015_

Round 1

Reviewer 1 Report

This seems like a very useful collection of equations presented in a very clear manner for nanomedicine researchers to take part of.

Author Response

Dear Reviewer,

Many thanks for your encouraging comment about our findings presented in the manuscript, where you said:

"This seems like a very useful collection of equations presented in a very clear manner for nanomedicine researchers to take part of."

Reviewer 2 Report

The manuscript technical note ID scipharm-1150758 report simple mathematical equations for the calculation of particle size and surface area of a various bioactive encapsulation systems.

The entry is well written and understandable, very general but in agreement with the requirement for a technical note.

I believe it could be interested for the readers who are working in different fields and managing delivery system, even if there is no new information, but it is a good summary.

I recommend adding some scheme related to the measurement of surface area and size to have a fast graphic approach. In addition, I suggest adding some examples of exception or not ordinary case example.

Author Response

We thank Reviewer for encouraging statements and valuable suggestions and comments. We particularly thank the encouraging statement:

"The manuscript technical note ID scipharm-1150758 report simple mathematical equations for the calculation of particle size and surface area of a various bioactive encapsulation systems.

The entry is well written and understandable, very general but in agreement with the requirement for a technical note."

With respect of Reviewer's suggestion:

"I recommend adding some scheme related to the measurement of surface area and size to have a fast graphic approach."

we have now added a new Figure to the manuscript a copy of each is attached here for the kind attention of Reviewer. A paragraph explaining this new image is now added to the article:

"While equation (6) can be used universally to measure surface area of all type of spherical drug carriers, quantification of particle number requires specific equations for different carrier systems based on their ingredients (and as a result their physicochemical properties) as explained throughout this article and simplified in Figure 3."

With respect of the other suggestion of the Reviewer:

"I suggest adding some examples of exception or not ordinary case example."

the following sentence is added to the beginning of last paragraph of section 2.2.:

"Exceptional cases for the use of equation (3) for quantification of particle number of vesicular drug carriers would be multilamellar vesicles (MLV) or multivesicular vesicles (MVV) (see Figure 2)."

We hope that the manuscript have addressed all Reviewer's comments and suggestions successfully. 

Sincerely

Prof. M. R. Mozafari (Corresponding Author)

------------------------------------------------------------

Reviewer 3 Report

How many particles are presented in the solution? It is an interesting yet important questions that worth to be investigated. M.R.Mozafari et al. present several simple equations to calculate the particle number of metallic, polymeric, and lipid nanoparticles. The manuscript is well written and easy to follow. However, I have some major concerns about it.

  1. Please define the Na in line 98. Please move the definition of ϱ, M, and D in equation 2 right next to the equation. Although authors defined these parameters after citing a reference, the readers might be confused whether it is applied for the equation given or just for the reference.
  2. Please explain why reference 25 (line 102-104) was cited here. From this paragraph, this reference was cited merely for the nanoparticle size. How it is related to equation 2 is uncertain.
  3. Figure 1 is a bit misleading. The authors didn’t specify the relationship between these different small molecules with the phosphatidylcholine backbone. Please add some details.
  4. I am confused about equation 3. I don't fully understand the rationale for this equation. If the authors can provide some references or a bit more introduction of how this equation evolves, it will be very beneficial.
  5. Please define the E in equation 4.

Author Response

Authors would like to thank this Reviewer for encouraging statement:

"... present several simple equations to calculate the particle number of metallic, polymeric, and lipid nanoparticles. The manuscript is well written and easy to follow. "

With respect of the Reviewer's suggestions and comments, we have gone through all of them carefully. Please find our responses after each comment or suggestion below:

  • Please define the Na in line 98.

The version of the formatted manuscript we are asked to perform changes on it does not have line numbers, hence we are not able to find "line 98". Therefore, we performed a search on the file (found no "Na" at all) and only found "NA" (Avogadro's number), which was already defined, and have highlighted it in yellow.

  • Please move the definition of ϱ, M, and D in equation 2 right next to the equation. Although authors defined these parameters after citing a reference, the readers might be confused whether it is applied for the equation given or just for the reference.

We completely agree with Reviewer and have done the suggested changes. 

  • Please explain why reference 25 (line 102-104) was cited here. From this paragraph, this reference was cited merely for the nanoparticle size. How it is related to equation 2 is uncertain.

A version of equation (2) is used in the article of Rajakumar et al. We have attached the full paper for the kind attention of Reviewer.

  • Figure 1 is a bit misleading. The authors didn’t specify the relationship between these different small molecules with the phosphatidylcholine backbone. Please add some details.

We have added more explanation to the Figure legend as suggested:

"Figure 1. Structural formula of phosphatidylcholine molecule and its derivatives / related phospholipids. The chemical moieties on the right panel replace the Choline moiety depicted on the left panel and hence each phospholipid molecule is named based on its specific moiety on its hydrophilic head group."

  • I am confused about equation 3. I don't fully understand the rationale for this equation. If the authors can provide some references or a bit more introduction of how this equation evolves, it will be very beneficial.

Equation (3):

NVes = [Ming × NA] / [Ntot × 1000]

is the main equation by which particle number of vesicular bioactive carrier systems can be calculated. A statement describing this point has now been added to the manuscript:

"Equation (3) is the main equation by which particle number of vesicular bioactive carrier systems can be easily calculated once we know Ntot. "

  • Please define the E in equation 4.

E stands for "exponent" and is now defined in the manuscript text. 

We hope that we have addressed all Reviewer's comments and suggestions. 

Sincerely

Prof. M. R. Mozafari (Corresponding Author)

-----------------------------------------------------------

Round 2

Reviewer 3 Report

The revised version has addressed all my concerns. It can be accepted.